# Microbiota-Accessible Borates as Novel and Emerging Prebiotics for Healthy Longevity: Current Research Trends and Perspectives

**DOI:** 10.3390/ph18060766

**Published:** 2025-05-22

**Authors:** Andrei Biţă, Ion Romulus Scorei, Marvin A. Soriano-Ursúa, George Dan Mogoşanu, Ionela Belu, Maria Viorica Ciocîlteu, Cristina Elena Biţă, Gabriela Rău, Cătălina Gabriela Pisoschi, Maria-Victoria Racu, Iurie Pinzaru, Alejandra Contreras-Ramos, Roxana Kostici, Johny Neamţu, Viorel Biciuşcă, Dan Ionuţ Gheonea

**Affiliations:** 1Drug Research Center, Faculty of Pharmacy, University of Medicine and Pharmacy of Craiova, 2 Petru Rareş Street, 200349 Craiova, Romania; andreibita@gmail.com (A.B.); george.mogosanu@umfcv.ro (G.D.M.); maria.ciocilteu@umfcv.ro (M.V.C.); gabriela.rau@umfcv.ro (G.R.); catalina.pisoschi@umfcv.ro (C.G.P.); roxana.kostici@umfcv.ro (R.K.); johny.neamtu@umfcv.ro (J.N.); 2Department of Pharmacognosy & Phytotherapy, Faculty of Pharmacy, University of Medicine and Pharmacy of Craiova, 2 Petru Rareş Street, 200349 Craiova, Romania; 3Department of Biochemistry, BioBoron Research Institute, S.C. Natural Research S.R.L., 31B Dunării Street, 207465 Podari, Romania; romulus_ion@yahoo.com; 4Department of Physiology, Escuela Superior de Medicina, Instituto Politécnico Nacional, Plan de San Luis y Diaz Mirón, Mexico City 11340, Mexico; msoriano@ipn.mx; 5Department of Pharmaceutical Technology, Faculty of Pharmacy, University of Medicine and Pharmacy of Craiova, 2 Petru Rareş Street, 200349 Craiova, Romania; 6Department of Instrumental and Analytical Chemistry, Faculty of Pharmacy, University of Medicine and Pharmacy of Craiova, 2 Petru Rareş Street, 200349 Craiova, Romania; 7Department of Rheumatology, Faculty of Medicine, University of Medicine and Pharmacy of Craiova, 2 Petru Rareş Street, 200349 Craiova, Romania; cristina.gofita@umfcv.ro; 8Department of Organic Chemistry, Faculty of Pharmacy, University of Medicine and Pharmacy of Craiova, 2 Petru Rareş Street, 200349 Craiova, Romania; 9Department of Biochemistry, Faculty of Pharmacy, University of Medicine and Pharmacy of Craiova, 2 Petru Rareş Street, 200349 Craiova, Romania; 10National Agency for Public Health, 67A Georghe Asachi Street, MD-2028 Chisinau, Moldova; maria-victoria.racu@ansp.gov.md (M.-V.R.); iurie.panzaru@ansp.gov.md (I.P.); 11Laboratory of Molecular Biology in the Congenital Malformations Unit, Children’s Hospital of Mexico Federico Gomez (HIMFG), Calle Dr. Marques 162, Col. Doctores, Alc. Cuahutémoc, Mexico City 06720, Mexico; acora_ramos@hotmail.com; 12Department of Toxicology, Faculty of Pharmacy, University of Medicine and Pharmacy of Craiova, 2 Petru Rareş Street, 200349 Craiova, Romania; 13Department of Physics, Faculty of Pharmacy, University of Medicine and Pharmacy of Craiova, 2 Petru Rareş Street, 200349 Craiova, Romania; 14Department of Internal Medicine–Pneumology, Faculty of Medicine, University of Medicine and Pharmacy of Craiova, 2 Petru Rareş Street, 200349 Craiova, Romania; biciuscaviorel@gmail.com; 15Department of Gastroenterology, Research Center of Gastroenterology and Hepatology, University of Medicine and Pharmacy of Craiova, 2 Petru Rareş Street, 200349 Craiova, Romania; dan.gheonea@umfcv.ro

**Keywords:** microbiota-accessible borates, prebiotics, nutrition, autoinducer-2–borate, host–microbiota symbiosis, healthy longevity

## Abstract

Precision nutrition-targeted gut microbiota (GM) may have therapeutic potential not only for age-related diseases but also for slowing the aging process and promoting longer healthspan. Recent studies have shown that restoring a healthy symbiosis of GM by counteracting dysbiosis (DYS) through precise nutritional intervention is becoming a major target for extending healthspan. Microbiota-accessible borate (MAB) complexes, such as boron (B)–pectins (rhamnogalacturonan–borate) and borate–phenolic esters (diester chlorogenoborate), have a significant impact on healthy host–microbiota symbiosis (HMS). The mechanism of action of MABs involves the biosynthesis of the autoinducer-2–borate (AI-2B) signaling molecule, B fortification of the mucus gel layer by the MABs diet, inhibition of pathogenic microbes, and reversal of GM DYS, strengthening the gut barrier structure, enhancing immunity, and promoting overall host health. In fact, the lack of MAB complexes in the human diet causes reduced levels of AI-2B in GM, inhibiting the *Firmicutes* phylum (the main butyrate-producing bacteria), with important effects on healthy HMS. It can now be argued that there is a relationship between MAB-rich intake, healthy HMS, host metabolic health, and longevity. This could influence the deployment of natural prebiotic B-based nutraceuticals targeting the colon in the future. Our review is based on the discovery that MAB diet is absolutely necessary for healthy HMS in humans, by reversing DYS and restoring eubiosis for longer healthspan.

## 1. Introduction

In healthy longevity, years of good health approximate the biological lifespan, with physical, cognitive, and social functioning that enables well-being. Lifespan is the total number of years a person lives, while healthspan is the number of years spent in good health. Improving health can lead to a longer lifespan. These terms are very well defined and can measure whether certain diets can provide human populations with a healthy life expectancy [1,2]. A person can have a long lifespan but a short period of health if they suffer from chronic diseases. It is believed that individuals with healthy longevity have a biological age that is younger than their chronological age [3,4].

Boron (B), an essential micronutrient for vascular plants [5], has recently been claimed to also be essential for the healthy symbiosis of the host microbiome [6].

B has a structural role in the cell walls by binding to pectin and is ubiquitous in higher plants, forming a complex with the rhamnogalacturonan-II (RG-II) of vegetable pectins crosslinked with B bridges [7]. The cell walls of plants contain up to 780 mg/kg of B, in proportion to the amount of apiose in the cell walls [8]. Pectic polysaccharides (pectins) are dietary fibers fermented mainly in the colon [9,10]. B forms bonds between two RG-II monomers to build B-dimerized RG-II, contributing to the formation of pectic substances. In plant cell walls, more than 90% of RG-II is borate dimerized under B-sufficient conditions. The most complex pectin, RG-II, accounts for only 10% of plant cellular pectin [11]. The commensal bacteria in the colon ferment dietary pectin to produce short-chain fatty acids (SCFAs), such as acetate, butyrate (BUT), propionate, and lactate, which help support healthy microbiota symbiosis [12,13].

Recently, a B compound of chlorogenic acids (CAs) (diester chlorogenoborate–DCB) [14,15] was identified in green coffee beans; phenolic acids are extremely important in plant metabolism and human nutrition [16]. DCB is an organic ester of boric acid (BA) with two molecules of CA monoesters (Figure 1), i.e., with the three most common mono-caffeoylquinic acids, namely 5-O-caffeoylquinic acid (5-CQA), 3-O-caffeoylquinic acid (3-CQA), and 4-O-caffeoylquinic acid (4-CQA).

B ester derivatives are used in dietary supplements, known for their contribution to a healthy lifestyle. Research has demonstrated their effectiveness in treating conditions such as osteoporosis and arthritis by effectively increasing the integration of calcium into bones, joints, and cartilage [17]. The essential attributes of these molecules are given by the indigestibility in the gastric environment of both B–pectin and B–phenols and the accessibility of these groups of organic B esters to the colonic microbiota [14,18].

There are examples of studies on the possible inclusion of phenols in the class of prebiotics [19,20].

For a food product to be classified as a prebiotic, the following criteria may be considered: (*i*) it must resist gastric acidity and enzymatic hydrolysis, (*ii*) it must not be absorbed into the upper gastric system, and (*iii*) it must be fermented by gut bacteria. There are numerous in vivo studies showing the relationship between the consumption of polyphenols and the health of the microbiota, which by their structure and resulting metabolites are substrates for probiotics, helping to increase commensal bacteria and reduce pathogenic bacteria, thus maintaining the health of the host–microbiota symbiosis (HMS) [21,22,23].

Gut flora-accessible borates or microbiota-accessible borates (MABs), such as B–pectic polysaccharides (RG–borate complexes; Figure 1) and B–phenolic esters (DCB complexes), are naturally occurring borates essential for HMS in longer healthspan. MABs mean non-digestible organic borates metabolized by microbes [18]. MABs are dietary components designed for limited digestion and absorption in the upper gastrointestinal tract, facilitating their significant delivery to the colon. Upon reaching the colon, they are metabolized by the gut microbiota (GM), which leads to the release of B and other bioactive moieties. Detailed information regarding their colonic availability, specific microbial enzymatic processing, and the subsequent absorption or excretion pathways—which constitute the absorption, distribution, metabolism, and excretion (ADME) profile for these dietary borate compounds—are further elucidated in key cited literature, including the patent application of Scorei et al. (2023) and recent comprehensive reviews on this topic [14,18]. MABs have special significance in precision nutrition for targeting GM in the future. By considering MABs in diet, it could influence the deployment of natural prebiotic B-based nutraceuticals targeting the colon to improve public health in the future. MABs are claimed to be essential for the biosynthesis of autoinducer-2 (AI-2) and have recently been proposed as new and emerging prebiotics [18,24,25].

This narrative review aims to consolidate current knowledge and emerging perspectives specifically focused on MABs as novel prebiotic candidates for promoting healthy HMS and longevity [26]. Given that the dedicated study of MABs—defined as particular organic borate complexes designed for colonic action—is a nascent and rapidly developing field, this review synthesizes foundational concepts, details their proposed unique mechanisms, discusses dietary implications, and explores the potential for healthy aging. Our approach emphasizes the specific contributions and understanding of these MABs, rather than providing an exhaustive systematic review of the broader, more established fields of general prebiotics or overall B biology, to highlight the distinct aspects of this emerging area of research.

The characterization of MABs as ‘innovative and emerging’ prebiotics is substantiated by proposed distinctive benefits and mechanisms compared to conventional prebiotics. A primary distinction lies in the direct action of the prebiotic B component delivered by MABs. This B is understood to be incorporated into the colonic mucus gel layer, leading to its enhanced impermeabilization. Such an effect is significant as it directly inhibits the adherence of pathogenic bacteria and hinders their ability to form biofilms on the mucus surface—a targeted mechanism for strengthening the gut barrier that is not typically associated with traditional prebiotics, which act predominantly through fermentation. Furthermore, another key differentiating factor is the potential for efficacy at lower daily intake levels. The minimum effective dose of indigestible B from MABs (e.g., at least 1 mg/day, derived from a maximum of approximately 1000 mg of MABs such as B–pectins or B–phenolic acids) is considered to be significantly less than the larger quantities of conventional fiber-based prebiotics often required to achieve broad gut health benefits. This suggests a potent and targeted mode of action for the B-mediated effects of MABs.

Élie Metchnikoff’s studies from more than a century ago argued that gut bacteria can cause aging and that the gut is the first organ to age [27,28]. It has been shown that gut aging is accompanied by dysbiosis (DYS) of the GM. Increased microbiota DYS causes systemic aging, either directly by triggering inflammation or through microbial compounds and thus increased cell senescence (aging). The intestine triggers a cascade of physiological processes that cause systemic aging of the intestine, and consequently, slowing the aging of the gut has the effect of slowing the aging of the whole body. Thus, a healthy microbiome can help prevent telomeres from wearing away. In combating intestinal aging, DYS is restored, with favorable effects for the distant organs of the human body, and the health of the whole body is improved [29,30].

While longer telomeres are correlated with a healthy diet and ideal body weight (body mass index (BMI), 25 kg/m^2^), short telomeres have been correlated with oxidative stress and increased inflammation [31]. Gut telomere-dependent aging controls whole-body aging. Combating intestinal aging and microbiome aging has the following results: (*i*) whole-body health is improved, (*ii*) gut DYS is restored, and distant organs of the human body can be restored. Furthermore, telomere shortening is correlated with lifestyle-related conditions such as high blood sugar levels, abdominal fat, and physical inactivity [32].

Studies have supported telomere length as an excellent biomarker of aging. And telomere shortening may be a molecular clock that triggers aging. In addition, telomere length decreases with aging, and age-related diseases accelerate this process.

Physiological processes that occur with aging include increased inflammation, oxidative stress, and a decrease in telomerase activity that causes accelerated telomere shortening [33].

In this review, we emphasize the need for a healthy diet, supplemented with MABs, to influence human healthy life expectancy through precision nutrition targeting the microbiome to prevent DYS, which affects the gut and is the first organ to age in the human body.

## 2. World’s Healthiest Eating Habits

In the following paragraphs, a compilation of data regarding the characteristics of diet linked to improved healthspan is presented. Some components, features, and reported changes in microbiota are highlighted, while relevant components such as B source are only mentioned for those where B-rich foods are key, since the B concentration of some foods has been previously measured and reviewed, and linked to some beneficial effects in human health [34,35,36,37].

### 2.1. Nordic Diet

The traditional Nordic diet (ND), also known as the Baltic Sea diet, is commonly consumed in Norway, Denmark, Finland, Sweden, Iceland, Greenland, and the Faroe and the Åland Islands. It is characterized by high-quality protein sources, complex carbohydrates, and nutrient-rich superfoods.

The ND is abundant in fruits and vegetables, including especially berries, cabbage, root vegetables, and legumes, as well as wild-gathered herbs and fresh herbs, mushrooms, nuts, potatoes, whole grains, and whole-grain cereals. It also incorporates rapeseed oil, oily fish (especially salmon, herring, and mackerel), and shellfish, along with seaweed, white and low-fat meat, and game. Additionally, the ND promotes the consumption of low-fat dairy products while discouraging the intake of sugar-sweetened products. Fish and rapeseed oil, which serve as the primary sources of *n*-3 polyunsaturated fatty acids (PUFAs), including α-linolenic acid, eicosapentaenoic acid, and docosahexaenoic acid, are key components of the ND. Moreover, salt consumption is restricted to 5–6 g/day, and vitamin D supplementation is advised as a dietary supplement [38].

People living in these countries primarily obtain their protein from fish and seafood, as well as dairy products like skyr, which is a fresh fermented milk cheese consumed similarly to yogurt. The ND is exceptionally high in omega-3 fatty acids (FAs), known for their anti-inflammatory properties and their role in supporting cardiovascular (CV) health. Additionally, skyr is an outstanding source of protein, making it a superfood. As a result, this food is highly beneficial for maintaining stable blood sugar levels. It also provides a good amount of calcium (Ca), which is essential for strong bones and teeth. Studies analyzing the microflora in traditional skyr cultures have identified *Lactobacillus delbrueckii* subsp. *bulgaricus*, *Streptococcus thermophilus*, and *L. helveticus* [39,40].

Recent research has indicated that the ND is associated with enhanced CV risk markers and a reduced likelihood of developing type 2 diabetes (T2D), colorectal cancer, stroke, inflammation, and all-cause mortality [41]. A pyramid represents the fundamental characteristics of the ND. At its base are the primary components of the ND, including northern vegetables, roots, cabbage, peas, northern fruits, apples, pears, and berries. The middle section contains whole-grain cereals (rye, oats, and barley), potatoes, legumes, nuts, and seeds. Positioned above these are oily fish, low-fat or fat-free dairy products, lean meat options (poultry and game), eggs, margarine, and rapeseed oil. At the top of the pyramid are foods that should be consumed in moderation, such as animal fats (butter), red and processed meat (beef, pork, processed meat products, and sausages), sweets, and chocolate [42].

In studies investigating the healthy ND, hippuric acid (HipA) was detected both in serum and in urine. HipA originates from the GM-mediated breakdown of dietary phenolic compounds. Specifically, HipA has been positively associated with *Faecalibacterium prausnitzii*, a microbial species that produces BUT [43].

The ND closely resembles the Mediterranean diet (MD), but it highlights foods from Nordic countries, along with plant-based foods and seafood. The key distinction is that it prioritizes canola oil over extra virgin olive oil.

### 2.2. Mediterranean Diet

The MD is traditionally rooted in the eating habits of countries bordering the Mediterranean Sea, particularly Greece, Southern Europe, and Southern Italy. It is characterized by a high intake of legumes, whole grains, vegetables, fruits, herbs, fish, and unsaturated fats such as olive oil, along with moderate consumption of red dairy products and wine, and a low intake of meat products.

Mediterranean-style eating has been shown to lower the risk of metabolic syndrome conditions, including T2D, high blood pressure (BP), and high cholesterol, all of which are risk factors for heart disease. It is also linked to longevity and maintaining a healthy weight. Additionally, the traditional MD is recognized for its brain-boosting benefits. A recent study found that individuals who adhered to this diet showed fewer hallmarks of Alzheimer’s disease (AD), with brain health comparable to that of individuals 18 years younger. Recent scientific evidence highlights that those who consume more fruits/nuts, vegetables, legumes, grains, and fish, while reducing their intake of dairy and meat/poultry, and consuming moderate amounts of ethanol, tend to have excellent CV and cognitive health [44].

The MD includes (*i*) daily consumption of unrefined cereals and related products (e.g., whole-meal pasta, whole-meal bread, and brown rice), along with vegetables, nuts, fresh fruit, and low-fat dairy products; (*ii*) olive oil as the primary source of lipids; (*iii*) moderate consumption of alcohol, preferably red wine, with meals; (*iv*) moderate intake of poultry, fish, eggs, potatoes, and sweets; (*v*) limited monthly consumption of red meat; and (*vi*) regular physical activity. It should be mentioned that the MD includes staple foods rich in B such as grape, broccoli, garlic, tomato, pomegranate, and olives combined with the consumption of drinking water with high levels of B, which frequently results in intakes of B higher than 13 mg/day [45].

A healthy gut has been linked to improved mental health, stronger immunity, and better stress management. The literature has demonstrated a positive correlation between adherence to the MD and the abundance of health-associated genes such as *Paraprevotella* spp. and *Bacteroides* spp. [46]. Adopting a sustainable MD can contribute to food security and healthy living for today’s population and future generations [47,48,49].

### 2.3. Green Mediterranean Diet

Scientists, dissatisfied with the traditional top-ranking MD, sought to explore its potential further. The Green MD (GMD) incorporates more plant-based foods, eliminates red meat, and increases polyphenol intake. While red meat is excluded, other sources of animal protein, such as dairy and fish, remain part of the diet. In studies examining its effects, the GMD specifically included a daily serving of 100 g of *Wolffia globosa* (Mankai), an aquatic flowering plant from the duckweed (*Lemnaceae*) family. Rich in protein, iron (Fe), vitamin B_12_, and prebiotic B, Mankai is served as a nearly equivalent meat substitute. This plant contains over 200 polyphenols and phenolic metabolites, exhibiting high antioxidant properties and abundant flavonoid-class polyphenols, such as luteolin and apigenin derivatives.

Green tea, another component, provides catechins, particularly epicatechin gallate, epigallocatechin, and epigallocatechin gallate polyphenols [50]. Additionally, urolithin A (UroA), an intestinal metabolite of ellagic acid found in Mankai and walnuts, has been linked to increased UroA concentrations. It can cross the blood–brain barrier (BBB), reduce neuroinflammation, enhance neurogenesis, and specifically protect the hippocampus from oxidative stress [51]. Similarly, tyrosol, primarily found in olive oil, has been shown to shield the hippocampus from neuroinflammation, stimulate neurogenesis, and improve spatial memory, while it and other molecules in olive oil protect against induced motor disruption [52,53].

The GMD includes a variety of polyphenol-rich foods such as other coffee, cocoa, teas, raisins, berries, red wine, herbs and spices, red onions, olive oil, olives, and flaxseed. It also influences the GM, with only a small fraction (5–10%) of ingested polyphenols absorbed in the upper gastrointestinal tract. The remaining polyphenols are metabolized by GM, producing microbiome-derived phenolic metabolites that enter the systemic circulation as bioactive compounds [54]. The GMD enhances *Prevotella* spp. degradation pathways (associated with vegetarian diets) while reducing *Bifidobacterium* spp. biosynthesis pathways in the GM [54], potentially benefiting cardiometabolic health regardless of weight loss [55]. A GMD, enriched with polyphenols and reduced in red meat consumption, could serve as an improved version of the MD [56].

### 2.4. Japanese and Korean Diet

Traditionally, the Japanese diet (J-diet) and Korean diet (K-diet) include a variety of fresh, unprocessed foods such as rice, fish, vegetables, and meat, along with gut-friendly fermented delicacies. While Korean cuisine is generally spicier, Japanese cuisine is known for its more subtle and delicate flavors. Japanese cuisine is often praised for its focus on fresh, whole ingredients and balanced flavors, whereas Korean cuisine also provides a balance of nutrients but tends to be higher in sodium (Na) and fat compared to Japanese cuisine.

Vegetables are a vital component of both J-diet and K-diet. Common Japanese vegetables include white radish, carrots, and spinach. “Kimchi”, fermented cabbage, is a fundamental part of Korean cuisine and adds a spicy kick and tangy flavor to many dishes. Spices play a more prominent role in Korean cuisine than in Japanese cuisine [57].

Both Japanese and Korean cuisine provide numerous health benefits [58] and incorporate a diverse range of plant-based foods, including seaweed, which is rich in iodine and supports hormonal health. In Japan, fermented soy products such as “natto”—fermented soybeans—are traditionally consumed. Natto is beneficial for gut health, packed with essential nutrients, and the best source of vitamin K_2_, which plays a crucial role in bone health. Additionally, the emphasis on fish as a protein source makes the diet abundant in omega-3 FAs. Green tea, a widely enjoyed beverage, is rich in antioxidants like polyphenols and catechins, which may contribute to brain and CV health.

Fermented foods are also a fundamental part of the K-diet [59], including “kimchi” and “jang”, which are commonly eaten alongside soups, seafood, rice, and meat. The Korean fermented food “kimchi” is particularly beneficial for gut health, providing a variety of live bacterial strains that nourish the microbiome. Research has also linked kimchi consumption to reduced risk factors for heart disease.

The concept of K-diet represents traditional Korean food culture, cooking methods, dietary habits, and food patterns, while K-foods refer to the specific food components of the K-diet, made exclusively with agricultural products grown in Korea. The key characteristics of the K-diet include (*i*) a variety of rice- and grain-based dishes; (*ii*) an emphasis on fermented foods; (*iii*) greater use of wild vegetables from both land and sea; (*iv*) higher consumption of legumes and fish with reduced intake of red meat; (*v*) frequent use of medicinal herbs like garlic, green onions, red peppers, and ginger; (*vi*) preference for sesame oil and Japanese basil oil (perilla); (*vii*) minimal deep-frying and use of fat in cooking; (*viii*) meals based on seasonal ingredients; (*ix*) a diversity of regional cuisines; (*x*) more home-cooked meals; (*xi*) all dishes prepared at temperatures below 100 °C, often blanched at 70–80 °C or lower, without the use of oil [60].

Both diets exhibit higher exposure to B compared to other diets, with beans, nuts, and seeds as the main source [61,62].

With an emphasis on plant-based foods and minimal processing, the inclusion of fermented foods makes both the J-diet and K-diet highly effective for promoting healthy GM.

### 2.5. Thai Diet

Rice and noodle dishes filled with colorful fruits and vegetables—some also featuring meat and seafood—are widely enjoyed in Thailand [63]. The foundation of the Thai diet is based on the five major food groups outlined in Thailand’s “Nutrition Flag”: rice and starch (including rice, bread, cereals, and pasta), vegetables, fruits, dairy (such as milk, yogurt, and cheese), and protein sources like meat (poultry), fish, dried beans, eggs, and nuts. The “Nutrition Flag” represents a balanced way of eating that enables Thais to maintain a “healthy diet” [64].

Thai cuisine is known for its bold flavors, enhanced by a variety of spices that offer numerous health benefits. Traditional Thai ingredients such as lemongrass, papaya, chili peppers, Thai basil, and turmeric are rich in antioxidants, plant compounds, and essential nutrients [65].

A staple ingredient in Thai soups and curries, coconut milk has been a subject of nutritional debate in recent years due to its high saturated fat content. However, one study found that regular and short-term consumption of coconut milk had no adverse effects on lipid profiles in the general population. In fact, it was associated with a reduction in low-density lipoprotein (LDL)–cholesterol—often referred to as “bad” cholesterol—and an increase in high-density lipoprotein (HDL)–cholesterol [66]. Additionally, fats such as coconut milk can promote satiety, reducing the likelihood of snacking between meals.

In Thailand, it is also common to begin the day with protein-rich breakfast options like eggs or pork, which not only enhance satiety but also help regulate blood sugar levels. However, fried Thai foods, such as “tempura”, are not the healthiest choice. Regular consumption of fried foods has been linked to an increased risk of CV diseases (CVDs) and higher all-cause mortality rates [67].

### 2.6. West African Diet

The West African diet is generally low in proteins and high in starchy carbohydrates such as yams, cassava, tubers, and rice [68]. People in Mali, Chad, Senegal, and Uruguay eat healthier foods than those in the US and many European countries [69]. These foods are an excellent source of energy and are often rich in nutrients. They contain vitamin B complex, which plays a role in many of the body’s key functions, including CV and cell health. The West African diet contains particularly nutrient-dense grains, such as sorghum, which is a good source of minerals, especially Fe, magnesium (Mg), and copper (Cu). While the traditional reliance on palm oil is potentially problematic, the spices used in almost every meal are a safe source of polyphenols, which have antioxidant and anti-inflammatory properties. These include garlic, melon seeds, and African nutmeg [70]. Currently, food consumption in some African households has shifted towards diets high in fats and oils, calorie-based sweeteners, and animal products high in saturated fats (diets commonly referred to as ‘Western diets’). Although some governments in African countries have established policies and programs to encourage a return to traditional diets, there is still no mainstream movement in favor of a return to the West African diet [71].

### 2.7. “Microbiome” Diet

The microbiome diet, created by Dr. Raphael Kellman, is based on consuming and avoiding specific foods with the goal of restoring gut health. It is also claimed to offer additional benefits, such as faster metabolism and weight loss. This three-phase program is designed to promote weight loss by improving gut health [72].

Phase 1 (21 days)–Gut reset. This initial and strictest phase focuses on eliminating unhealthy bacteria from the gut while restoring stomach acids and digestive enzymes. It also aims to repair the gut lining by incorporating prebiotics and probiotics. The key components of this phase include the following: (eliminate)—removing harmful foods, toxins, and chemicals that may cause inflammation or disrupt gut bacteria, including pesticides, hormones, antibiotics, and certain drugs; (repair)—consuming plant-based foods and supplements that support gut healing and microbiome balance; (replace)—adding specific herbs, spices, and supplements to restore stomach acid, digestive enzymes, and improve gut bacteria quality; (restore microbiota)—reintroducing beneficial bacteria through probiotic- and prebiotic-rich foods [72].

During this phase, a wide range of foods are restricted, including all grains, eggs, most legumes and dairy, starchy fruits and vegetables, processed and fried foods, sugar, artificial sweeteners, certain fats, fish, and meat. Instead, the diet encourages consuming organic, plant-based foods and prebiotic-rich foods like asparagus, garlic, onions, and leeks. Additionally, fermented foods such as sauerkraut, “kimchi”, kefir, and yogurt are recommended for their probiotic content. Certain supplements—including probiotics, zinc (Zn), vitamin D, berberine, grapefruit seed extract, wormwood, and oregano oil—are also strongly encouraged [72].

Phase 2 (28 days)–Metabolic stimulation. This phase aims to further strengthen the gut and microbiome while allowing more dietary flexibility. While foods that were restricted in Phase 1 should still be avoided 90% of the time, there is some leeway—up to four weekly meals can include previously excluded foods [72].

Additionally, dairy, free-range eggs, gluten-free grains, and legumes can be gradually reintroduced. Over time, more fruits and vegetables, including mangoes, melons, peaches, pears, sweet potatoes, and yams, can also be added back into the diet [72].

Phase 3–Lifelong maintenance. The final phase is intended for long-term maintenance and does not have a specific duration. It is designed to help maintain weight loss and sustain gut health. This phase should be followed until the desired weight loss goal is achieved and aims to support long-term well-being [72].

The “microbiome” diet primarily promotes the intake of non-starchy fruits and vegetables, fermented foods, grass-fed meats, and low-mercury (Hg) wild-caught fish. Beyond specific dietary guidelines, it includes additional lifestyle recommendations. For instance, it advocates for consuming organic foods and avoiding chemicals found in unnatural household cleaners and personal care products. The use of a high-quality water filter is also encouraged. This diet is believed to enhance gut health by minimizing exposure to toxins, pesticides, and hormones [72].

Furthermore, it recommends various supplements to help reduce inflammation, eliminate harmful bacteria, and support gut health. Examples include Zn, glutamine, berberine, caprylic acid, quercetin, garlic, grapefruit seed extract, wormwood, oregano oil, probiotics, and vitamin D. Additionally, it advises against excessive use of certain medications—such as antibiotics, non-steroidal anti-inflammatory drugs (NSAIDs), and proton pump inhibitors—that may disrupt gut bacteria balance [73,74].

The “microbiome” diet is rich in probiotics and prebiotics while being low in added sugar, all of which contribute to a healthier gut. It also warns against the overuse of specific medications that could negatively affect gut health. This diet may offer additional benefits for older adults, mainly by encouraging the consumption of fruits, vegetables, healthy fats, lean proteins, and other plant-based foods. It also restricts sugary, fried, and processed foods, prioritizing fresh produce, lean protein, healthy fats, and foods high in probiotics and prebiotics [75].

### 2.8. Dietary Approaches to Stop Hypertension Diet

The dietary approaches to stopping hypertension (DASH) are a flexible and balanced eating plan designed to promote a heart-healthy lifestyle for life [76]. The DASH diet emphasizes foods rich in minerals such as potassium (K), Ca, and Mg, along with protein and dietary fiber. It focuses on vegetables, fruits, and whole grains while incorporating fat-free or low-fat dairy products, fish, poultry, beans, and nuts. The diet restricts foods high in salt (Na) and also limits added sugar and saturated fat, such as those found in fatty meats and full-fat dairy products. The standard DASH diet allows up to 2300 mg of salt per day, whereas a lower Na version reduces Na intake to 1000 mg daily [77,78].

Research on East Asian populations has also examined the connection between the DASH diet and T2D. One prospective study reported that higher adherence to a DASH-style diet was linked to a 29% lower risk of developing T2D [79]. Another intervention study investigated the glycemic effects of a DASH-style diet and found notable reductions in glycated hemoglobin (HbA1C) in the DASH-style diet group compared to other intervention groups [80]. Additionally, the DASH diet proved effective in lowering BP and other CVD risk factors, including blood glucose, blood lipids, body weight, and waist circumference [81].

A key principle of the diet is increasing the intake of nutrient-rich foods known for their role in lowering BP. The DASH diet promotes consuming fruits, vegetables, whole grains, lean proteins, and low-fat dairy products while reducing Na and sugar in beverages and processed foods. By following these dietary guidelines, individuals are encouraged to achieve and maintain optimal BP levels. The combination of the DASH diet and reduced sodium intake demonstrated complementary effects in decreasing bone turnover, leading to improved bone mineral status. This was evidenced by reductions in serum osteocalcin, C-terminal telopeptide of type I collagen (CTX), serum parathyroid hormone (PTH), and urinary Ca and uric acid levels [82]. Moreover, multiple studies have consistently shown an association between the DASH diet and lower all-cause mortality rates [83,84].

### 2.9. Food Restriction as Key Factor in Longevity

Dietary restriction has a broader definition, encompassing the limitation of a specific nutrient (macro or micro) or periodic feeding restriction (fasting). Under controlled conditions, food restriction does not cause malnutrition but can lead to significant metabolic changes over time. This occurs because nutrient restriction triggers the redistribution of macronutrients, decreasing the presence of nutrients that may be detrimental in certain diseases or health conditions [85].

A calorie-restricted diet (hypocaloric diet) primarily involves an overall reduction in total calorie intake from all macronutrients. However, this reduction can be categorized as either a ‘low-calorie diet’ or a ‘very low-calorie diet’, with the latter defined as consuming fewer than 800 kcal. Such a low intake is strongly linked to severe nutritional deficiencies, increasing metabolic risks such as undernutrition, and is contraindicated for children and young people. A ‘low calorie diet’, in contrast, has a moderate reduction (up to 40% of total calories) that continues to provide sufficient nutrients for whole-body functioning, and is normally in the form of a calorie restriction or hypocaloric diet [86].

Recently, dietary restriction has been gaining prominence at the scientific level [87], encompassing both caloric restriction and the limitation of certain nutrients that support pathogenic bacteria. These nutrients fall into the following categories: (*i*) those that require restriction for a healthy and long life, such as sulfur (S), Fe, and gluten; and (*ii*) those that should be abundant in the diet to promote longevity and well-being, including prebiotic B, omega-3 FAs, polyphenols, SCFAs (mainly BUT), medium-chain fatty acids (MCFAs; mainly caproic acid), and probiotic foods obtained through fermentation (yogurts, cheese, pickles) [18].

### 2.10. Intermittent Fasting

Various intermittent fasting approaches exist, including limiting food intake to specific hours, known as time-restricted eating (TRE). This method typically involves a daily eating window of 8–10 h, lasting anywhere from 4 to 12 weeks, with some studies requiring adherence for only 5 out of 7 days per week. Nearly all research on TRE indicates weight loss, reduced adiposity, and improvements in circulating factors associated with CVDs [88].

Intermittent fasting has been shown to induce significant changes in GM content and composition, leading to an increase in the *Firmicutes*/*Bacteroidetes ratio*, which helps reverse age-related changes in GM. Periodic use of a fasting-mimicking diet (FMD), which consists of a plant-based, low-calorie, low-protein five-day dietary intervention followed by a normal diet, has beneficial effects on both cellular function and overall health. Human studies suggest that chronically reducing calorie intake by 15–20% below normal levels has profound effects on risk factors for T2D, cancer, and CVDs. Moreover, increased levels or activity of certain hormones, factors, and genetic pathways influenced by proteins, specific amino acids, or sugars have been consistently linked to accelerated aging and/or age-related diseases in organisms from yeast to humans [89].

Undergoing three cycles of FMD has been found to reduce body weight, trunk and total body fat, and BP, while also lowering insulin-like growth factor-1 (IGF-1) without causing serious adverse effects. Studies show that IGF-1 levels are significantly higher in individuals from the US consuming a high-protein diet compared to those following a low-protein diet. Recent scientific findings also highlight the link between healthy gut symbiosis and reduced IGF-1 levels [90]. Commensal gut bacteria such as *Lactobacillus* spp. and *Bifidobacterium* spp. are associated with BUT production and decreased IGF-1 in humans. In contrast, Western diets contribute to elevated insulin levels, hyperglycemia, high IGF-1, cholesterol, and triglycerides, which not only activate pro-aging pathways but also promote insulin resistance, obesity, and various age-related diseases. The combination of these factors appears to accelerate aging while also increasing morbidity and mortality, both by hastening the aging process and by fostering diseases independent of aging [91,92].

Table 1 summarizes the main characteristics and food groups traditionally used in the world’s healthiest eating habits.

## 3. Microbiome and Exceptional Longevity

Intestinal aging has been shown to be accompanied by DYS of the GM. Increased GM DYS will cause systemic aging directly through bacterial components or by triggering inflammation and increasing cellular senescence [31].

The gut is the tip of the cascade of events that initiate systemic aging. The gut is crucial to health and plays a role in aging, and slowing the aging of the gut slows the aging of the whole body. The shortening of telomere length is linked to physiological oxidative and inflammatory responses modulated by the composition of the GM. Thus, it has been hypothesized that a healthy HMS may help prevent telomere attrition [30].

Aging relates to inflammation and deoxyribonucleic acid (DNA) damage. By combating intestinal aging, the health of the entire body is improved, and DYS is restored with favorable effects for the distant organs of the human body [94].

Short telomeres are associated with DNA damage and are first noticed in the intestine. Shortened telomeres have been detected in the intestinal epithelium of patients with intestinal inflammation, and it has been stated that intestinal aging is telomere-dependent and controls the aging of the entire human body [95].

Studies on the consumption of certain foods have been shown to be correlated with healthy GM: the consumption of nuts, seeds, vegetables, and fruits. People consuming higher amounts of nuts, seeds, fruits, and vegetables (the foods richest in MABs) have been associated with longer telomeres than people who consume mainly meat and who have a shorter telomere length, associated with pathogenic bacteria [96]. There are many studies showings that B modulates the human microbiome and that nutrition rich in prebiotic B slows aging by mitigating oxidative and inflammatory stress [6,97,98]. It was found in northern France that a B-rich diet was inversely correlated with mortality [99].

Prebiotic B intake in the daily diet has been shown to attenuate pathologies associated with aging such as cognitive decline, cancer, sarcopenia, and bone health. These findings have shown that an intake of MABs (prebiotic B) promotes a longer healthspan [100]. We know that all foods of vegetable origin contain prebiotic B in different amounts depending mainly on the genetics of the plant and the ecosystem [18].

In addition, the gut of centenarians is known to retain high levels of BUT and an increased content of BUT-producing bacteria (BPB) [101]. Gut DYS in the elderly increases the risk of diseases associated with aging. GM composition changes with age, causing mild inflammation in the elderly. Combating gut aging and the aging of the microbiome yields the following results: (*i*) whole-body health is improved; (*ii*) gut DYS is restored, and the removed organs of the human body can also be realized [102]. Clinical models prove how correcting age-related gut DYS of the elderly is beneficial and justify nutritional approaches that consider the HMS to combat aging and its associated diseases.

BPB are mainly a group of commensals (non-pathogenic) bacteria. BPB are mainly anaerobic and can ferment carbohydrates and polyphenols in the colon. The *Firmicutes* phylum of bacteria includes the main producers of BUT in the healthy colon. The GM of long-living humans (centenarians) has been enriched with more BPB and genes, which is beneficial for reducing the occurrence of neurodegenerative diseases and maintaining host health. These results suggest that BPB or BUT play an important role in longer healthspan for humans [103]. In the gut, BPB are inversely correlated with the aging process: BUT production in the gut lumen is vital for maintaining the gut mucosal barrier. All animal rejuvenation procedures result in a significant increase in the relative abundance of BPB. In addition, the GM of young people contains more BPB than that of the elderly [104]. BUT produced by commensal bacteria is an essential inhibitor of unhealthy aging. BUT prevents inflammation, insulin resistance, colon cancer, and cognitive decline; supports repair and reconstruction in situations such as strokes; prevents damage to the spine; and mitigates accelerated aging [105].

The production of BUT by commensal microbes is reduced when humans age and by the bacteria fermentation of dietary fiber in the lower gastric system (colon) stimulating the production of a pro-longevity hormone called fibroblast growth factor 21 (FGF21), which plays an important role in regulating the body’s energy and metabolism.

BUT raises the level of brain-derived neurotrophic factor (BDNF) and the number of neurons; triggers the adenosine monophosphate (AMP)-activated protein kinase (AMPK) pathway, a guardian enzyme of mitochondrial metabolism and homeostasis; and produces a high expression of sirtuin 1 (SIRT1), a silent regulatory protein of information in the liver, and an enzyme with a role in oxidative stress, which is specific to increasing the longevity of living beings. BUT production in the intestinal lumen is vital for maintaining the intestinal mucosal barrier [106,107].

All animal rejuvenation procedures result in a significant increase in the relative abundance of BPB [108]. In addition, the GM of young people contains more BPB than that of the elderly [101]. BPB are negatively correlated with animal protein intake: a diet high in animal protein induces a reduction in BPB and may lead to a favorable environment for pathogenic bacteria.

Reducing animal protein intake increases longer healthspan, decreases frailty, reduces body weight and body fat, increases physical performance, increases glucose tolerance, and improves essential metabolic markers [109,110]. The AI-2 signaling molecule of the bacterial quorum favors BPB [111]. Numerous recent studies prove that the AI-2 molecule limits pathogenic microbes and restores eubiosis after antibiotic treatment [112].

AI-2, a bacterial quorum-sensing (QS) signaling molecule, enhances colonization by BPB [102]: recent in vivo studies support the protective role of AI-2 against pathogens and the restoration of normal microbiota composition after antibiotic treatment [112]. Moreover, an increased level of AI-2 in microbiota results in the proliferation of BPB that were depleted by antibiotic treatment [113]. Consequently, the AI-2 signaling molecule may enter a feedback loop and rescue DYS and intestinal inflammation [114].

## 4. Microbiota-Accessible Boron Complexes to Longer Healthspan

Regarding human clinical evidence for MABs as a nutritional strategy promoting healthy longevity, current research is progressive. While extensive randomized controlled trials or large cohort studies specifically linking MAB consumption to a longer healthspan or diminished age-related diseases are part of ongoing and future research, initial human data offer insights. Notably, a randomized pilot study assessing dietary a ‘MABs-cocktail’ reported improvements in aging biomarkers in participants after 60 days of supplementation, suggesting a potential contribution to healthy lifespan extension. This preliminary human evidence is supported by various in vitro and in vivo (animal) studies, such as those detailed by Scorei et al.’s (2025) Provisional Patent Application [26], which elucidate beneficial mechanisms of MAB components, including antioxidant effects and modulation of gut health indicators pertinent to longevity.

Microbiota is required for improved biological age compared to chronological age. An increased chronological age does not indicate healthy aging. In numerous studies, it has been reported that targeting GM can help both for age-related diseases and for slowing aging in humans for longer healthspan [113,115,116].

An important feature of aging is low-grade inflammation due to GM DYS, which accelerates aging in humans [31]. Gut DYS is connected with many pathologies, such as malnutrition, obesity, T2D, hepatic steatosis, metabolic syndrome, chronic hepatitis, irritable bowel syndrome, CVDs, autoimmune diseases, neurological disorders, psychiatric disorders, neurodegenerative diseases, renal diseases, respiratory diseases, sarcopenia, cancer, autism spectrum disorders, sleep disorders, insomnia, allergies, and infections. Therefore, the prevention of age-related GM DYS may be favorable to longevity [117,118].

It is known that levels of reactive oxygen species due to inflammation can cause the *Firmicutes* bacterial phylum found in abundance in the colon to become inactive. The *Firmicutes* phylum contains more than 200 bacterial species and is the majority in the commensal microbiome [111]. Recent scientific evidence shows that as commensal microbiota decreases, the abundance of proinflammatory pathogenic bacteria proliferates and causes age-related diseases and increased mortality in humans [23,119].

Older humans, compared to younger humans, have increased *Bacteroidetes* bacterial phylum, and a significant positive correlation with mortality risk has been shown to be independent of age. In addition, the aging of humans shows a decrease in the *Firmicutes*/*Bacteroidetes* ratio and a proliferation of pathogenic bacteria and correlated with a decrease in the abundance of BPB useful in the storage of the bacteria [120,121]. In addition, aging is a factor that induces DYS. Gut DYS in the elderly increases the risk of age-related diseases.

The restoration of age-dependent GM DYS is a future therapeutic target to fight to stop aging and its related diseases [121,122].

New scientific data on B prove that (*i*) B is required for a healthy symbiosis of commensal microbes in humans; (*ii*) B is not essential for cellular metabolism but is only required for a healthy symbiosis in the gut, mouth, scalp, vagina, and skin; and (*iii*) naturally occurring MAB complexes are promising new prebiotic candidates, while inorganic B compounds (BA and borate salts) are indigestible and, in some cases, may be toxic [6,15,18]. Indigestible organic borates (borate esters of CA and isochlorogenic acid) and plant cell wall B–pectins (RG-II) fermented by the GM have been defined as MABs (see Figure 1 for chemical structures) [18,27]. MABs are required for a healthy symbiosis of commensal bacteria and the human host and are emerging prebiotic agents targeting the colon [18].

Pectins, a soluble fiber found mainly in apples, citrus fruits, and berries, are recognized for promoting intestinal health, acting as a prebiotic and stimulating the growth of beneficial bacteria. Due to the complex nature of pectic polysaccharides, they can support distinct fermentation characteristics and can promote the selective growth of different intestinal microbiota species [123]. CA has also demonstrated the potential to act as a ‘prebiotic’ in the human large intestine [124]. The presence of esterified B in the structure of pectins and phenols, during their metabolism by commensal bacteria producing AI-2, leads, through enzymatic metabolism, to the release of free borate anions in the colon at basic pH, which chemically react to form a complex with 4,5-dihydroxy-2,3-pentanedione (DPD; pro-AI-2 signaling molecule), resulting in AI-2–borate (AI-2B; a B-containing QS signaling molecule) [18,125].

MAB species are involved in the generation of AI-2B signaling molecules, the inhibition of the proliferation of pathogenic microbes, strengthening the colonic mucus gel for an integral intestinal barrier, and enhancing healthy human host immunity (Figure 2). AI-2 is a well-conserved QS signal, synthesized by a large cohort of Gram-negative and Gram-positive bacteria, and has the capacity to mediate both intra- and inter-species communication. Antibiotic-induced intestinal DYS can be partially counterbalanced by artificially increasing AI-2 levels [114]. Surprisingly, a large number of bacteria that respond robustly to AI-2 do not encode LuxP- or LsrB-type AI-2 receptors, which raises the possibility of the existence of other unrecognized receptor types for this autoinducer. For example, in bacterial pathogens such as *Pseudomonas aeruginosa* and *Enterococcus faecalis*, gene expression and phenotypes such as biofilm formation and virulence factor production are regulated by AI-2. AI-2 is not a single signaling molecule, but a group of DPD derivatives that can rapidly interconvert. Two AI-2 forms engaged by corresponding bacterial receptors have been identified, including the B-containing DPD derivative (2*S*,4*S*)-2-methyl-2,3,3,4-tetrahydroxytetrahydrofuran-borate ((*S*)-THMF-borate), recognized by LuxP-type 2 receptors, and the non-borated (2*R*,4*S*)-2-methyl-2,3,3,4-tetrahydroxytetrahydrofuran ((*R*)-THMF), recognized by LsrB receptors found in enteric bacteria and in some members of several other families. The two forms of AI-2, (*S*)-THMF-borate and (*R*)-THMF, can interconvert, and the addition of borate is known to shift the equilibrium of AI-2 molecules towards the (*S)*-THMF-borate form. The presence of borate anions in the bacterial medium led to a 119-fold decrease in the specific binding affinity of non-borated AI-2. A reduction in AI-2 leads to a proliferation of *Firmicutes* bacteria, in particular. The AI-2 level in anaerobic digestion is mainly governed by *Firmicutes* [126]. The inhibition of pathogenic bacteria is driven by the increase in the level of borated AI-2 and the proliferation of AI-2-producing bacteria.

The intestinal mucus layer serves as an interface between the host’s internal environment and the complex community of GM. The mucus layer acts as a physical shield and modulates immune responses by selectively allowing the passage of molecules and signaling compounds that facilitate immune surveillance and regulate inflammation. Recent studies show that the *Firmicutes* phylum is more abundant in the mucus gel layer of the colon in humans and rats compared to the *Bacteroidetes* phylum [127,128].

Reduced consumption of foods rich in MABs causes a decrease in the content of AI-2B in the intestinal microbiota, which is a decrease in the proliferation of bacteria from the *Firmicutes* phylum, the main producers of BUT, with negative effects on the health of the host microbiome [18].

Dysbiotic gut amplifies aging with major effects on the erosion of genetic potential in humans and animals and leads to decreased proliferation of bacteria of the *Firmicutes* phylum, AI-2, and implicitly AI-2B [18]. The *Firmicutes* phylum, which has over 200 bacterial species in its composition, mainly contributes to the production of BUT [111].

Deficiency in the consumption of MABs will lead to the DYS of the microbiome, the mucus gel will degrade, and the proliferation of pathogenic microbes will be stimulated with direct negative effects on the musculoskeletal system and a decrease in the immunity of the human body.

Recent studies have shown that bacteria from the *Firmicutes* phylum, by proliferating, increase the concentration of AI-2 and BUT, an essential biomarker in longer healthspan [129]. BUT is considered an essential inhibitor of unhealthy aging and physiological decline through the following main actions: it sources energy for the healthy state of colonocytes, prevents gut inflammation, modulates adipogenesis, and, in particular, strengthens the gut barrier function against the penetration of bacteria into the body’s bloodstream.

It was thus demonstrated that AI-2 induced the proliferation of the *Firmicutes* phylum when the microbiome was treated with antibiotics, by inhibiting the proliferation of the *Bacteroidetes* phylum and reversing antibiotic-induced DYS [130].

The BPB belonging to the *Firmicutes* phylum were the following most important bacterial species: *Butyrivibrio fibrisolvens*, *Clostridium butyricum*, *C. kluyveri*, *Eubacterium limosum*, and *F. prausnitzii*. Moreover, other bacterial phyla also produce BUT by processing lactate and acetate resulting from glucose metabolism, such as *Anaerostipes* spp., *Bifidobacterium* spp., and *E. hallii* [111].

BPB are predominantly anaerobic and proliferate in the colon of young people and are more abundant in rural than urban people. Moreover, it has been shown that people with good longevity have a high proportion of the following bacterial genera: *Clostridium* cluster XIVa, *Christensenellaceae* and *Ruminococcaceae* (*Firmicutes*), *Akkermansia* (*Verrucomicrobiota*) [18].

MABs have been shown to be required for a healthy HMS, and the consumption of prebiotic B attenuates many age-related diseases such as cancer, cognitive decline, and sarcopenia and promotes bone health. Consequently, the consumption of MABs in a diet promotes longer healthspan, has the ability to reverse microbiome DYS, is associated with enhanced immunity and musculoskeletal health [100,131,132], and may in the future become a diet for longer healthspan.

Prebiotic MABs have the following actions: (*i*) ensure a healthy HMS, (*ii*) strengthen colonic mucus gel with B species through the transfer of the borate anion to the mucins of the mucus gel layer, (*iii*) inhibit pathogenic bacteria from adhering to the mucus gel and inhibit the proliferation of pathogenic microbes, and (*iv*) stimulate the proliferation of BPB and reverse GM DYS. Moreover, there is a link between nutrition rich in MABs, healthy HMS, healthy host metabolism, and longer healthspan [7,18].

An example of the protective role of commensal microbes against pathogenic bacteria is as follows: AI-2B produced from *Ruminococcus obeum* can confuse *Vibrio cholerae*, resulting in the premature repression of AI-2B QS-mediated virulence and decreased colonization in the intestine [133]. We conducted experiments on rats that show the relationship between AI-2B and fecal BUT levels and an increase in the *Firmicutes* phylum [26]. Furthermore, there are studies on ruminants that show an increase in fecal BUT concentration after B ingestion [134].

## 5. Microbiota-Accessible Borate Diet: A Targeted and Precise Nutrition

The B of MABs is required for AI-2 and, implicitly, AI-2B synthesis and it affects the behavior of GM bacteria. AI-2B is formed from AI-2 in the presence of prebiotic B (organic B arrived in the colon undigested both as borate–phenolic acids and as B–pectin) in the plant composition [7,15]. Consumption of MABs in the daily diet is necessary for the restoration of the microbiome, which in B-prebiotic deficiency causes the DYS and degradation of colonic gel mucus and mucus in the oral cavity [18].

Ingestion of MABs daily leads to the arrival of prebiotic B as complexes of pectins and phenolic acids in the colon, where it reacts with DPD (pro-AI-2 signaling molecule), generating AI-2B molecules and has the following effects: (*i*) it stimulates the proliferation of BPB by decreasing the concentration of AI-2; (*ii*) it transfers the borate anion from BUT-stimulated mucin; (*iii*) it inactivates the mucus structure of the colonic gel and increases its impermeability to commensal and pathogenic bacteria; and (*iv*) it inhibits the proliferation of pathogenic bacteria by increasing the level of AI-2B, where a high level of AI-2B protects against GM DYS [18].

The consumption of products with a high concentration of prebiotic B complex (vegetables, fruits, seeds, oceanic fish, fermented) [18] provides the following effects (Figure 3): (*i*) restoration of colonic mucus gel with far-reaching effects on the musculoskeletal and immune systems; (*ii*) rejuvenation of the microbiota and mucus gel of the colon; (*iii*) protects against GM DYS, where the reversal of DYS slows host aging [18]; (*iv*) stimulation of the proliferation of BPB in the colon; (*v*) inhibition of the proliferation of pathogenic microbes [7]; and (*vi*) protective diet against periodontal diseases [135].

BUT is the primary metabolite of MAB fermentation in the colon, and this is the major SCFA produced by BPB, associated predominantly with the bacterial *Firmicutes* phylum, and thus, prebiotic B is claimed to promote the rejuvenation of the microbiota–gut system [18].

In the context of MAB intervention, ‘microbiota rejuvenation’ refers to a documented shift in the GM composition from a profile often associated with aging or DYS towards one more characteristic of healthier, younger individuals. Key indicators of this rejuvenation include improvements in microbial diversity, a beneficial modulation of the *Firmicutes*/*Bacteroidetes* ratio (often reflecting a reversal of age-related changes), an increased abundance of beneficial commensal bacteria such as BPB within the *Firmicutes* phylum [25], and a reduction in pathobionts. These compositional and functional shifts are typically assessed using standard microbiome analysis methodologies, such as 16S ribosomal ribonucleic acid (rRNA) gene sequencing or shotgun metagenomics.

For MAB-personalized nutrition, in our previous paper, boron nutrient density (BND; mg/1000 calories) was defined as the ratio of nutrient content to the total energy content of food [18]. In general, for plant foods, about 10% of the total BND (BDN_T_) is accessible to microbiota and therefore indigestible. This essential accessibility to microbiota BND (BND_AM_) varies from one food to another and is increased in polyphenol-rich foods and, in particular, in foods rich in phenolic acids. The percentage (%BND_AM_) of 10% of the BND_T_ for vegetables is calculated from the quantitative ratio between B–carbohydrates and B–polyphenols (on average, 10:1); the indigestibility of B–carbohydrates is, on average, 1%, while B–polyphenols have an average indigestibility of 90%. For example, the 10 mg of BND_T_ needed to have a minimum of 1 mg of B in the stools is provided by a 90% diet rich in fruits, vegetables, seeds, fermented foods, and marine fish [18].

The fundamental feature of the MAB diet is its B level, expressed as BDN_T_, which should be provided from a minimum B intake of 10 mg B/day. The tolerable upper intake level (UL) is 0.16 mg/kg body weight/day, which is equivalent to a UL of 10 mg/person/day in adults. This UL also applies to pregnant and lactating women. These UL values apply only to the intake of B in the form of BA and borates [18,136]. This MAB diet (2000 kcal/day for a 70 kg person) is composed of 90% vegetables, fruits, seeds, fermented foods, and ocean fish and should be low in iron, sulfur, and gliadins and rich in polyphenols and omega-3 FAs (Table 2). The level of B in stool will be a minimum m of 1 mg/day [18].

## 6. Trends and Perspectives: Essentiality of Boron in Host–Microbiome Symbiosis and Targeted Precision Nutrition

New perspectives on the importance of B for humans and animals suggest that a new concept may be necessary to understand the role of B in health. Recent studies have demonstrated the role of B in the healthy symbiosis between commensal microbes [6,14,18,24]. Many studies are needed before B can be declared essential in nutrition; however, scientific data support the idea of the essentiality of this element in the healthy symbiosis between the microbiota and the human host, both through the presence of the borated AI-2 signaling molecule and through the presence of B in the mucus gel layer of the colon [6,137].

An accurate and targeted nutrition plan focuses on a specific health goal and tailoring diets to specific health needs. Precise and targeted nutrition refers to the practice of customizing food plans based on individual health needs, genetic predispositions, and lifestyle factors. Targeted and precise nutrition ensures that the body receives the right balance of nutrients, improving absorption and utilization. Personalized eating plans can help manage and prevent chronic diseases by addressing specific risk factors. Improving gut health through targeted nutrition can also have an impact on mental health, thanks to the gut–brain axis [138]. Without a doubt, this concerns the benefits of some described diets (like those found above in this text, several clearly include B as MABs in the included food).

Targeted and precise nutrition [139] in healthy longevity, according to the concept of the prebiotic B diet (MAB diet) has the following fundamental features:

(*i*) Impermeabilization of the mucus gel layer through the pathogen and commensal bacteria [6];

(*ii*) Inhibition of pathogenic bacteria [18,27];

(*iii*) Stimulation of BPB in the colon [18].

The effects of this supplementation with MABs as DCB are mainly the following:

(*i*) Rewilding the human GM (by increasing BPB) [6,18];

(*ii*) Intestinal rejuvenation through the accumulation of B in the mucus gel layer [15] and an increase in cell apoptosis, and promotion of intestinal cell proliferation [140].

MABs consist of B–pectins (RG–borate) and borate–phenolic esters (DCB). The B content of these complexes is high, depending on the B supply of the soil. Therefore, the nutritional density of B is an important nutritional indicator that needs to be known [18]. The differences are primarily determined by the B requirement for the saturation of the mucus gel layer, which inhibits the attachment and proliferation of pathogenic bacteria to it. It is already known that phenolic acids have inhibited the growth of pathogenic bacteria such as *Escherichia coli*, *Staphylococcus aureus*, *Salmonella typhimurium*, *L. rhamnosus*, *C. perfringens*, *C. difficile*, and *Listeria monocytogenes* [141].

The goal of this diet is to have a longer healthspan and increase healthy life expectancy. In the future, the need for prebiotic B supplementation (functional foods with high-B content, prebiotic B dietary supplements such as DCB and MABs diet) will become strictly essential for the healthy symbiosis and mitigation of age-related DYS for a healthy lifespan in humans.

Further considering the unique contributions of MABs to gut health and longevity necessitates a comparative understanding against established prebiotics. While conventional prebiotics, such as inulin and fructooligosaccharides (FOS), are well recognized for their capacity to be selectively fermented by beneficial GM, thereby promoting their growth and the production of SCFAs, MABs are proposed to operate through additional, distinct B-dependent pathways. These include the modulation of inter-species bacterial communication via AI-2B signaling and the direct structural fortification of the colonic mucus gel layer through B incorporation. To rigorously assess the relative efficiencies and potential synergistic interactions, future research should prioritize direct comparative studies. Such investigations would be invaluable for precisely defining the specific advantages or complementary roles of MABs in fostering beneficial bacterial populations, enhancing gut barrier integrity, and improving overall health outcomes relative to, or in combination with, other prebiotic strategies.

Future research work is desirable to improve specific benefits to those adopting a B-enriched diet (like that described for microbiota and gut–senescence modulation in this text); in particular, efforts should be centered on personalizing the supplements not only to compare the benefits of supplementation from food or synthesized compounds but also to establish which compound is the best to induce the best benefits in elongated healthspan depending of the characteristic of the targeted population.

## Figures and Tables

**Figure 1 pharmaceuticals-18-00766-f001:**
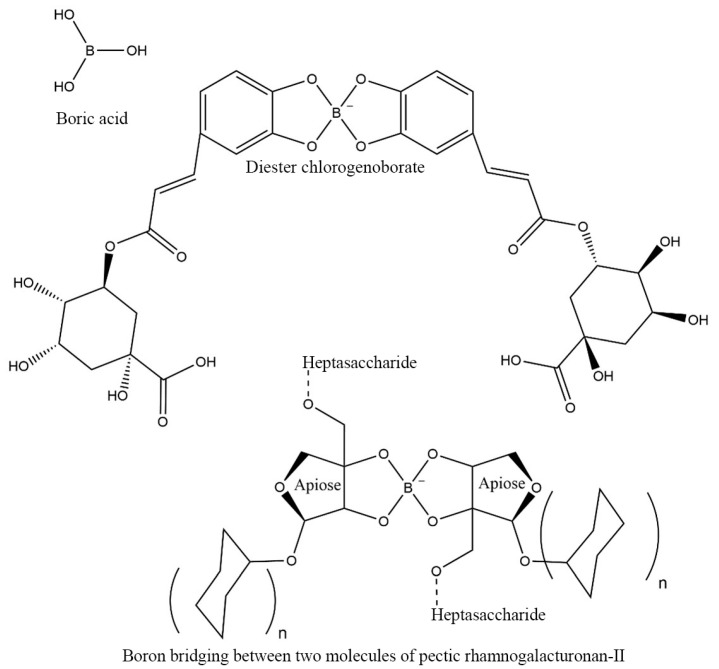
Chemical structure of boric acid and two naturally occurred borates (potential microbiota-accessible borates). The induced changes in microbiota after its administration seem linked to improved healthspan. Diester chlorogenoborate represents borate phenolic esters, while the complex with boron between two-apiose molecules belongs to the borate–pectic polysaccharides.

**Figure 2 pharmaceuticals-18-00766-f002:**
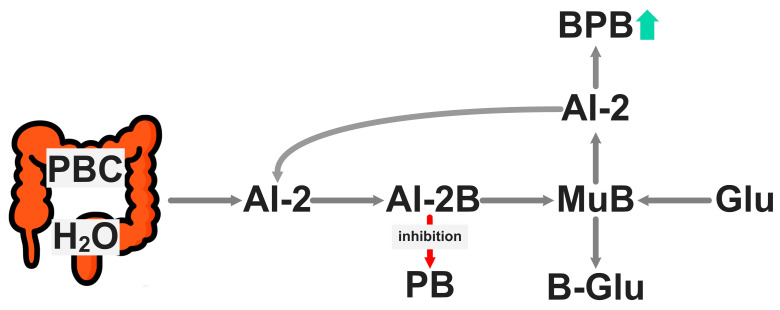
Mechanism of action involving AI-2B in the gel layer of colonic mucus in humans. AI-2: Autoinducer-2; AI-2B: Autoinducer-2–borate; B-Glu: Monosaccharide–boron complex; BPB: Butyrate-producing bacteria (mainly *Firmicutes* phylum); Glu: Monosaccharide (mainly fucose and sialic acid); MuB: Mucin gel–borate complex; PB: bacterial pathogen; PBC: prebiotic boron complex.

**Figure 3 pharmaceuticals-18-00766-f003:**
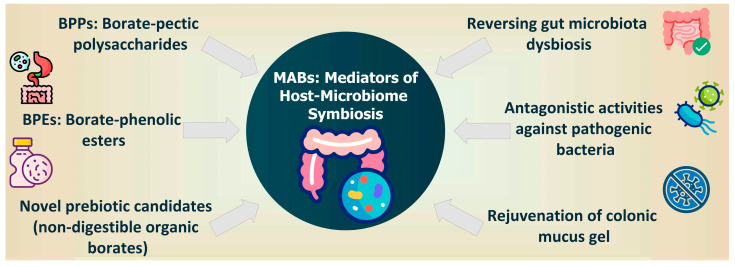
MABs as novel and emerging prebiotics for personalized nutrition: trends and perspectives.

**Table 1 pharmaceuticals-18-00766-t001:** The main food groups that are traditionally used for the healthiest dietary patterns in the world.

Dietary Pattern	Main Features	References
Nordic diet	▪ vegetables and fruits (cabbage, berries, root vegetables and legumes), wild-gathered herbs and mushrooms, potatoes, fresh herbs, nuts, whole-grain cereals, whole grains, oily fish (salmon, herring, mackerel), rapeseed oil, shellfish, seaweed, white and low-fat meat, low-fat dairy products, game (Nordic Nutrition Recommendations 2023 suggest 10 mg/day of B); ▪ avoiding products sweetened with sugar.	[38,39,40,93]
Mediterranean diet	▪ whole grains and unprocessed cereals (such as whole wheat bread, whole wheat pasta, and brown rice), along with fresh fruits, vegetables, nuts, and low-fat dairy products; ▪ olive oil as the primary fat source; ▪ moderate intake of alcohol, ideally red wine, consumed with meals; ▪ balanced consumption of fish, poultry, potatoes, eggs, and sweets; ▪ limited intake of red meat, typically on a monthly basis; ▪ engaging in regular physical activity.	[44,46]
Green Mediterranean diet	▪ more plant-based foods and more polyphenols (Mankai plant, green tea, coffee, cocoa, berries, raisins, red wine, red onions, herbs and spices, olives, olive oil, flaxseed); ▪ sources of animal protein, such as dairy and fish; ▪ no red meat.	[50,51,64]
Japanese and Korean diet	▪ various rice- and grain-based recipes; ▪ more fermented foods (“natto”, “kimchi”, “jang”); ▪ more vegetables from wild landscapes and the seas; ▪ more legumes and fish; ▪ more medicinal herbs (green tea, garlic, green onions, red peppers, ginger); ▪ more sesame oil and Japanese basil oil (perilla); ▪ more home-cooked meals (below 100 °C without using oil); ▪ less red meat and limited deep-fried cooking with fat.	[57,58,59,60]
Thai diet	▪ rice and starch (rice, bread, cereals, pasta); ▪ vegetables, fruits, dried beans, and nuts; ▪ dairy (milk, yogurt, cheese, coconut milk); ▪ meat (poultry, fish, eggs) and seafood; ▪ spices (lemongrass, papaya, chili peppers, Thai basil, turmeric).	[64,65,77]
West African diet	▪ low content of proteins; ▪ high content of starchy carbohydrates (yams, cassava, tubers, rice, sorghum); ▪ spices (garlic, melon seeds, African nutmeg).	[68,70]
“Microbiome” diet	▪ organic, plant-based diet and foods rich in prebiotics (asparagus, garlic, onions, leeks); ▪ fermented foods rich in probiotics (sauerkraut, “kimchi”, kefir, yogurt); ▪ certain supplements (probiotics, Zn, vitamin D, berberine, grapefruit seed extract, wormwood, oregano oil); ▪ dairy, free-range eggs, gluten-free grains, and legumes; ▪ non-starchy fruits and vegetables (mangoes, melons, peaches, pears, sweet potatoes, yams); ▪ grass-fed meats and low-Hg wild-caught fish.	[72]
DASH diet	▪ vegetables, fruits, and whole grains; ▪ fat-free or low-fat dairy products, fish, poultry, beans, and nuts; ▪ less foods that are high in salt, added sugar, and saturated fat (such as in fatty meats and full-fat dairy products).	[77,78]
Calorie restricted diet	▪ reduced consumption of prebiotic B, omega-3 FAs, polyphenols, SCFAs (primarily BUT), MCFAs (such as caproic acid), and probiotic-rich fermented foods (including yogurt, cheese, and pickles); ▪ limiting certain nutrients (such as sulfur, iron, and gluten) for better health and longevity.	[86,87]
Intermittent fasting	▪ daily eating window of 8–10 h, varying in duration from 4 to 12 weeks; ▪ plant-based, low-calorie, low-protein five-day dietary intervention.	[88,89]

BUT: Butyrate; DASH: dietary approaches to stop hypertension; FAs: fatty acids; MCFAs: medium-chain fatty acids; SCFAs: short-chain fatty acids.

**Table 2 pharmaceuticals-18-00766-t002:** The calories, B content, and total and microbiota-accessible BND (averages) for MAB-personalized nutrition.

Food Group	Calories/100 g	B Content (mg/100 g)	BND_T_ (mg B/1000 Calories)	BND_AM_ (mg B/1000 Calories)
Fruits	40	0.5	12.5	1.25
Vegetables	30	0.3	10	1
Seeds	600	1.5	2.5	0.25
Fermented foods	70	0.15	2.1	0.21
Marine fish	150	0.12	0.8	0.08

B: boron; BND: boron nutrient density; BND_AM_: accessible to microbiota BND; BND_T_: total BND.

## Data Availability

The original contributions presented in this study are included in the article. Further inquiries can be directed to the corresponding author.

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
