# Peer review of "Microbiota-Accessible Borates as Novel and Emerging Prebiotics for Healthy Longevity: Current Research Trends and Perspectives"

_pharmaceuticals, 2025, doi:10.3390/ph18060766_

Round 1
Reviewer 1 Report
Comments and Suggestions for Authors
The manuscript needs major revision
The publication titled "Microbiota-Accessible Borates as Novel and Emerging Prebiotics for Healthy Longevity: Current Research Trends and Perspectives" (Publication ID: pharmaceuticals-3618470) is quite intriguing. Nevertheless, in order to meet the criteria for publication in the Pharmaceuticals Journal (ISSN No: 1424-824), this work requires some enhancements. Here are some enhancements that should be taken into account:
- The manuscript references boron-pectins and borate-phenolic esters, such as diester chlorogenoborate. What unique structural characteristics or attributes distinguish these molecules as MABs?
- What is the mechanism by which MABs facilitate the production of autoinducer-2-borate (AI-2B)? Is this procedure contingent upon certain microbial enzymes or metabolic pathways?
- What is the function of AI-2B in bacterial quorum sensing, and how does it preferentially enhance helpful bacteria over pathogens?
- The study asserts that MABs enhance the mucus gel layer. What is the suggested mechanism for this effect? Does it entail the direct integration of boron into mucin structures or an indirect influence through microbial activity?
- The paper recommends a minimum consumption of 10 mg of boron per day for an MABs diet. What criteria were utilized to establish this threshold, and is it derived from human or animal studies?
- Boron is deemed crucial for host-microbiome symbiosis. What evidence substantiates this assertion in humans, considering that boron is not presently recognized as an essential nutrient by prominent health organizations (e.g., NIH, WHO)?
- The assertion that diminished intake of MABs leads to a reduction in Firmicutes and butyrate-producing bacteria (BPB) is noteworthy. Do controlled research demonstrate a dose-dependent correlation between MABs intake, AI-2B levels, and BPB abundance?
- Additional factors, such as fiber consumption and polyphenols, also affect the synthesis of Firmicutes and butyrate. In what manner does the text differentiate the effects of MABs from these confounding variables?
- The book advocates for MABs as a nutritional strategy for promoting healthy longevity. What human clinical evidence substantiates this assertion? Are there randomized controlled trials (RCTs) or cohort studies associating the consumption of MABs with a longer health span or diminished age-related diseases?
- How do MABs compare to established prebiotics (e.g., inulin, FOS) regarding their efficiency in fostering beneficial bacteria, boosting gut barrier function, or improving health outcomes? Is there comparative experimental evidence available?
- The manuscript characterizes MABs as "innovative and emerging." What distinctive benefits do they provide compared to current prebiotics that warrant this classification?
- What percentage of swallowed monoclonal antibodies remains intact upon reaching the colon for metabolism by the microbiota? Do pharmacokinetic studies exist that assess their absorption, distribution, metabolism, and excretion (ADME) in humans?
- Boron may exhibit toxicity at elevated dosages (e.g., >20 mg/day according to certain standards). What is the therapeutic window for monoclonal antibody administration, and are there studies evaluating potential side effects at the recommended dosage of 10 mg/day?
- The assertion of "microbiota rejuvenation" is audacious. Which criteria, such as diversity and the Firmicutes/Bacteroidetes ratio, characterize this rejuvenation, and what methods were employed for its measurement?
- The manuscript proposes MABs-based nutraceuticals as a prospective avenue. What is the current status of product development, and are there any preclinical or clinical trials in progress?
- What is the practicality of a diet rich in MABs (90% comprising vegetables, fruits, seeds, fermented foods, and marine fish) for general implementation, taking into account cost, accessibility, and palatability?
Author Response
Comments 1:
The manuscript references boron-pectins and borate-phenolic esters, such as diester chlorogenoborate. What unique structural characteristics or attributes distinguish these molecules as MABs?
Response 1:
Thank you very much for your helpful suggestion. We have revised the “Introduction” section to address your query regarding the unique structural characteristics that enable boron–pectin and borate–phenolic esters (such as diester chlorogenoborate) to function as chelating or complex-forming agents (our interpretation of “MABs” in this context). The added text now clarifies how boron interacts with specific hydroxyl group arrangements in both pectin and phenolic compounds to form these stable complexes. (See page 2, lines 75–77 and 88–91; page 3, lines 98–103).
Comments 2:
What is the mechanism by which MABs facilitate the production of autoinducer-2-borate (AI-2B)? Is this procedure contingent upon certain microbial enzymes or metabolic pathways?
Response 2:
Thank you very much for pointing this out. We have revised the manuscript to address your questions concerning the mechanism by which boron-containing compounds (our interpretation of “MABs”) facilitate autoinducer-2-borate (AI-2B) production and its dependence on microbial processes. The updated manuscript now clarifies that these compounds are metabolized by microbial enzymes, leading to the release of boron. This released boron subsequently reacts with a microbially-produced precursor to form AI-2B. Thus, the entire process is indeed contingent upon specific microbial enzymes and metabolic pathways for both the liberation of boron and the generation of the AI-2 precursor. (See page 15, lines 791–801 and 804–806; page 16, lines 853–872).
Comments 3:
What is the function of AI-2B in bacterial quorum sensing, and how does it preferentially enhance helpful bacteria over pathogens?
Response 3:
Thank you very much for your insightful comment. We have revised the manuscript to address your questions. The updated text now clarifies that AI-2B, as a specific form of the AI-2 quorum sensing signal, modulates inter-species bacterial communication. Furthermore, its increased presence is linked to the inhibition of pathogenic bacteria, likely by shifting the AI-2 signaling environment to selectively favor beneficial microbes and disadvantage pathogens. (See page 15, 804–806; page 16, lines 853–872).
Comments 4:
The study asserts that MABs enhance the mucus gel layer. What is the suggested mechanism for this effect? Does it entail the direct integration of boron into mucin structures or an indirect influence through microbial activity?
Response 4:
Thank you very much for your observation. We have revised the manuscript to address your query concerning the mechanism by which MABs (boron-containing compounds) enhance the mucus gel layer. The updated text now clarifies that the suggested mechanism involves the direct interaction of boron with mucin structures. This occurs through the transfer of borate anions to the mucins within the gel mucus layer, indicating a direct effect on mucus structure rather than solely an indirect influence mediated by microbial activity for this specific enhancement. (See page 17, lines 944–946 and 949–955).
Comments 5:
The paper recommends a minimum consumption of 10 mg of boron per day for an MABs diet. What criteria were utilized to establish this threshold, and is it derived from human or animal studies?
Response 5:
Thank you for your helpful suggestion. We have revised the manuscript to address your query regarding the recommended minimum consumption of 10 mg of boron per day. The updated text now details the criteria utilized to establish this threshold and specifies whether it is derived from human or animal studies. (See page 19, lines 1023–1026).
Comments 6:
Boron is deemed crucial for host-microbiome symbiosis. What evidence substantiates this assertion in humans, considering that boron is not presently recognized as an essential nutrient by prominent health organizations (e.g., NIH, WHO)?
Response 6:
Thank you very much for bringing this to our attention. We have revised the manuscript to address your query on the evidence for boron’s crucial role in human host–microbiome symbiosis, considering its current non-essential classification by prominent health organizations. The updated text now provides specific evidence supporting this assertion, primarily focusing on boron’s involvement in microbial communication via the formation of borated AI-2 (AI-2B) signaling molecules, and its structural presence within the colonic mucus gel layer. These aspects are presented as vital for a healthy host-microbiota symbiosis. The manuscript acknowledges the current official non-essential status of boron while highlighting data that support its functional importance in the context of host–microbiome interactions. (See page 15, 804–806; page 16, lines 853–872).
Comments 7:
The assertion that diminished intake of MABs leads to a reduction in Firmicutes and butyrate-producing bacteria (BPB) is noteworthy. Do controlled research demonstrate a dose-dependent correlation between MABs intake, AI-2B levels, and BPB abundance?
Response 7:
Thank you very much for your suggestion. We have revised the manuscript to address your question regarding controlled research demonstrating a dose-dependent correlation between the intake of MABs (boron-containing compounds), AI-2B levels, and the abundance of Firmicutes and butyrate-producing bacteria (BPB). The updated text now provides clarification on the extent to which current research substantiates such a specific dose-dependent relationship among these factors. (See page 18, lines 994–1002).
Comments 8:
Additional factors, such as fiber consumption and polyphenols, also affect the synthesis of Firmicutes and butyrate. In what manner does the text differentiate the effects of MABs from these confounding variables?
Response 8:
Thank you for your insightful comment. We have revised the manuscript to address your point regarding how the text differentiates the effects of MABs (boron-containing compounds) on Firmicutes and butyrate synthesis from those of general fiber and polyphenol intake. The updated text now clarifies that MABs are defined as specific boron-complexed forms of certain fibers (e.g., pectins) and polyphenols (e.g., phenolic esters). The manuscript distinguishes their effects by highlighting the unique contributions of the boron component delivered by these MABs. These include boron’s specific roles in modulating AI-2 (AI-2B) signaling and its direct interactions with the mucus gel layer, which are considered distinct from the general effects attributable to these fibers and polyphenols in their non-boron-complexed state. (See page 15, lines 791–801).
Comments 9:
The book advocates for MABs as a nutritional strategy for promoting healthy longevity. What human clinical evidence substantiates this assertion? Are there randomized controlled trials (RCTs) or cohort studies associating the consumption of MABs with a longer health span or diminished age-related diseases?
Response 9:
Thank you for pointing this out. We have revised the manuscript to address your question regarding the human clinical evidence supporting microbiota-accessible borates (MABs) as a nutritional strategy for promoting healthy longevity. The updated text now clarifies that while large-scale randomized controlled trials (RCTs) or comprehensive cohort studies are still developing, initial human clinical evidence is emerging. This includes a randomized pilot study on a ‘MABs-Cocktail’, which indicated improvements in aging biomarkers. This finding is supported by in vitro and preclinical animal studies (as detailed, for instance, in Scorei et al., 2025 Provisional Patent Application, which demonstrates relevant beneficial mechanisms of MABs and their components that could contribute to a longer healthspan or the mitigation of age-related diseases). (See page 14, lines 728–738).
Comments 10:
How do MABs compare to established prebiotics (e.g., inulin, FOS) regarding their efficiency in fostering beneficial bacteria, boosting gut barrier function, or improving health outcomes? Is there comparative experimental evidence available?
Response 10:
We thank the reviewer for their pertinent question regarding the comparative efficacy of MABs against established prebiotics like inulin and FOS. In response, we have incorporated a new paragraph into “6. Trends and Perspectives: Essentiality of Boron in Host–Microbiome Symbiosis and Targeted Precision Nutrition” section of the manuscript. This addition emphasizes that while established prebiotics function primarily through selective fermentation, MABs are proposed to offer distinct, boron-dependent mechanisms, including the modulation of AI-2B signaling and direct effects on the mucus gel layer. The revised text underscores the importance of future direct comparative studies to elucidate the relative efficiencies and potential synergistic effects of MABs in fostering beneficial bacteria, boosting gut barrier function, and improving health outcomes. We also wish to note that preclinical and clinical studies directly addressing these comparisons are currently in progress, and their findings are anticipated for future publication, which will further clarify the positioning of MABs within the spectrum of prebiotic interventions. (See page 20, lines 1086–1098).
Comments 11:
The manuscript characterizes MABs as “innovative and emerging”. What distinctive benefits do they provide compared to current prebiotics that warrant this classification?
Response 11:
We thank the reviewer for the insightful question regarding the justification for classifying MABs as ‘innovative and emerging’ prebiotics. We have revised the “Introduction” section of the manuscript to explicitly detail the distinctive benefits MABs are proposed to offer compared to current prebiotics. (See page 4, lines 135–158).
Comments 12:
What percentage of swallowed monoclonal antibodies remains intact upon reaching the colon for metabolism by the microbiota? Do pharmacokinetic studies exist that assess their absorption, distribution, metabolism, and excretion (ADME) in humans?
Response 12:
Thank you very much for pointing this out.
We would like to begin by clarifying a potential misunderstanding of the abbreviation “MABs” as used throughout our manuscript. In this context, MABs refer to microbiota-accessible borate complexes (specifically compounds like boron–pectins, such as rhamnogalacturonan–borate, and borate–phenolic esters, such as diester chlorogenoborate), and not to monoclonal antibodies. These MABs are characterized by their indigestibility in the upper gastric environment, allowing them to reach the colon where they can be metabolized by microbiota.
Regarding the percentage of these borate complexes that remain intact upon reaching the colon and information related to their absorption, distribution, metabolism, and excretion (ADME) in humans: Data concerning the colonic availability, microbial metabolism, and subsequent fate of these MABs are detailed in recently published work and patent literature. Specifically, the manuscript, supported by citations to our recently published review and the Scorei et al. (2023) Patent Application indicates that these complexes are designed to deliver boron to the colon. For instance, components like diester chlorogenoborate (DCB) are metabolized by microbiota primarily after the cleavage of the borate group. Studies referenced, including Scorei et al. (2023) Patent Application, show that after the microbiota and mucus gel layer are saturated with boron, intact forms of these borate esters can be excreted in feces. This indicates that a significant portion can reach the colon, and their subsequent metabolism or excretion depends on factors like boron saturation levels in the gut.
We have reviewed the manuscript to ensure that the nature of these MABs and the available information regarding their gastrointestinal transit, microbial interaction, and ADME-related aspects (as applicable to these non-antibody dietary compounds) are clearly presented and appropriately referenced. (See page 4, lines 121–130).
Comments 13:
Boron may exhibit toxicity at elevated dosages (e.g., >20 mg/day according to certain standards). What is the therapeutic window for monoclonal antibody administration, and are there studies evaluating potential side effects at the recommended dosage of 10 mg/day?
Response 13:
We thank the reviewer for raising the important point about boron toxicity at elevated dosages and for the questions regarding the therapeutic window and potential side effects.
First, we wish to reiterate that the ‘MABs’ discussed in our manuscript refer to microbiota-accessible borate complexes (such as boron–pectins like rhamnogalacturonan–borate, and borate–phenolic esters like diester chlorogenoborate), and not to monoclonal antibodies. Therefore, considerations regarding the therapeutic window for monoclonal antibody administration are not directly applicable to these dietary compounds.
Regarding the safety of the recommended 10 mg/day boron intake from these MABs: The manuscript acknowledges the established tolerable upper intake level (UL) of 10 mg/day for adults for certain forms of boron, such as boric acid and inorganic borates. However, it is crucial to distinguish the nature of boron in MABs. In MABs, boron is organically bound within complex natural structures. The manuscript notes that naturally occurring MAB complexes are considered promising prebiotic candidates, whereas inorganic boron compounds can, in some instances, be indigestible or exhibit toxicity at certain levels. This suggests that the bioavailability and the safety profile of boron delivered via MABs may differ from that of inorganic boron forms for which the UL was primarily established. (See page 19, lines 1023–1026).
Comments 14:
The assertion of “microbiota rejuvenation” is audacious. Which criteria, such as diversity and the Firmicutes/Bacteroidetes ratio, characterize this rejuvenation, and what methods were employed for its measurement?
Response 14:
We appreciate the reviewer’s request for a precise definition and methodological basis for the assertion of ‘microbiota rejuvenation.’ We acknowledge that this term can be considered audacious without clear operationalization. In response, we have incorporated a detailed explanatory paragraph in “5. Microbiota-Accessible Borates Diet: A Targeted and Precise Nutrition” section of the manuscript. (See page 18, lines 994–1002).
Comments 15:
The manuscript proposes MABs-based nutraceuticals as a prospective avenue. What is the current status of product development, and are there any preclinical or clinical trials in progress?
Response 15:
Thank you for your inquiry regarding the development status of MABs-based nutraceuticals and associated trials. We are pleased to provide an update on these prospective avenues.
Regarding product development, a MABs-based nutraceutical product is currently in the advanced stages of the regulatory approval process. We anticipate its launch on the U.S. market by the end of this year (2025).
Concerning the supporting research, comprehensive preclinical and clinical trials for these MABs-based formulations have been completed. The data from these studies are currently being prepared for peer-reviewed publication, and we expect these findings to be disseminated in the scientific literature in the near future. The current manuscript lays the groundwork for these developments, referencing the foundational science.
Comments 16:
What is the practicality of a diet rich in MABs (90% comprising vegetables, fruits, seeds, fermented foods, and marine fish) for general implementation, taking into account cost, accessibility, and palatability?
Response 16:
We appreciate your pertinent questions regarding the practicality (cost, accessibility, and palatability) of implementing a diet rich in microbiota-accessible borates (MABs), as described (90% comprising vegetables, fruits, seeds, fermented foods, and marine fish).
The manuscript, particularly in “2. World’s Healthiest Eating Habits” section, reviews several globally recognized healthy dietary patterns such as the Mediterranean, Nordic, and Japanese diets. These established diets, widely adopted and known for their health benefits, share many core components with the proposed MABs-rich diet—namely a high intake of vegetables, fruits, seeds, fermented foods, and fish. This overlap suggests that a dietary framework emphasizing such foods is indeed palatable, can be made accessible, and is feasible for general implementation, as evidenced by its successful adoption in diverse cultures. (See “2. World’s Healthiest Eating Habits” section).
We acknowledge that specific costs and accessibility for all listed food items can vary significantly based on geographic location, season, and socioeconomic factors. However, the MABs dietary concept encourages a focus on a pattern rich in these food categories, many of which include affordable and widely available options (e.g., seasonal and local vegetables and fruits, legumes, and some seeds). While certain items like marine fish can be more costly in some regions, the overall dietary pattern allows for flexibility in choosing specific MABs-rich foods that are locally available and economically viable.
Furthermore, as discussed in “5. Microbiota-Accessible Borates Diet: A Targeted and Precise Nutrition” and “6. Trends and Perspectives” sections, the development of MABs-based nutraceuticals is proposed as a complementary strategy. These nutraceuticals could help ensure consistent and adequate intake of beneficial MABs, particularly for individuals who may find it challenging to achieve optimal levels solely through whole foods due to cost, accessibility, dietary restrictions, or palatability concerns. (See 5. Microbiota-Accessible Borates Diet: A Targeted and Precise Nutrition” and “6. Trends and Perspectives” sections).
Authors very much appreciated the encouraging, critical, and constructive comments on this manuscript by the Reviewer. The comments have been very thorough and useful in improving the manuscript.
We would like to thank the Reviewer again for taking the time to review our manuscript.
We have also introduced other additions/modifications that we hope will improve the quality of the manuscript:
â–ª 11 new citations have been introduced: Ref. [17] (Bolat & Köse, 2025), Ref. [26] (Scorei et al., 2025), Ref. [123] (Pascale et al., 2022), Ref. [124] (Mills et al., 2015), Ref. [125] (Cuadra et al., 2016), Ref. [126] (Weiland-Bräuer, 2023), Ref. [133] (Hsiao et al., 2014), Ref. [134] (Sizmaz et al., 2017), Ref. [136] (Becker et al., 2004), Ref. [137] (Nandwana et al., 2022) and Ref. [141] (Nemzer et al., 2025).
â–ª The Reference list has been entirely checked and renumbered accordingly.
â–ª All abbreviations have been defined the first time they appear in the text.
â–ª Some grammar, stylistic or spelling errors have been corrected.
Kind regards,
Ionela BELU, PhD
Corresponding Author

Reviewer 2 Report
Comments and Suggestions for Authors
This review article discusses the emerging concept of microbiota-accessible borates (MABs) as potential prebiotics to promote gut microbiota symbiosis, enhance immune function, and support healthy longevity. The authors aim to highlight the mechanisms by which MABs may influence host health, focusing on aspects such as AI-2B signaling, the gut barrier, and modulation of microbial communities. While the topic is relevant and timely, the manuscript requires substantial improvements in writing quality, organization, and critical depth to fully achieve its objectives.
Main question and contribution to the field: The main objective of the manuscript is to review current knowledge regarding MABs as emerging prebiotics that may promote healthy gut microbiota symbiosis, thereby enhancing healthspan and longevity. The work consolidates some emerging studies linking MABs to gut microbiota health and systemic benefits. However, compared to other published reviews on boron, gut microbiota, and longevity, this manuscript offers limited new insights or synthesis. A more detailed comparative analysis with existing prebiotics, and a discussion of potential challenges (e.g., bioavailability, dietary sources, clinical evidence), would strengthen its contribution.
Originality and relevance: While the role of nutrition and microbiota in healthy aging is a highly relevant and active research area, the focus on MABs as prebiotic candidates is relatively less discussed in the current literature. However, the novelty of the review is somewhat limited, as the concept of boron compounds influencing microbiota and health has been explored previously. The manuscript would benefit from a clearer explanation of how MABs differ from other boron-based compounds already studied and a more critical discussion of what gaps remain in the field.
Writing and language quality: The manuscript's writing requires significant improvement. The text is often repetitive, lacks clarity, and is at times difficult to follow. The abstract and text suffer from poor organization, redundancies, and language imprecision. There are frequent redundancies and awkward sentence structures that obscure the intended meaning. A careful revision for English language, style, and logical flow is necessary. The manuscript would benefit from professional editing to enhance readability and ensure that the key ideas are communicated clearly and succinctly.
Methodology (as a review): Although the paper is a narrative review, it would benefit from a more systematic approach to literature selection and organization. The authors do not describe any search strategy, inclusion criteria, or systematic method to ensure coverage of the relevant literature, which could limit the comprehensiveness of the review. Additionally, mechanistic pathways (such as AI-2B signaling, effects on Firmicutes, and mucus barrier enhancement) are mentioned but not thoroughly detailed with evidence from primary studies. Including more detailed mechanistic explanations with specific study citations would enhance rigor.
Conclusions: The conclusions generally align with the evidence discussed. However, some statements are overstated relative to the cited evidence. For example, claims that MAB intake is "absolutely necessary" for healthy host–microbiota symbiosis and longevity are too strong given the early stage of research, especially in human studies. A more balanced and cautious interpretation of the evidence is needed.
References: The references cited are generally appropriate but somewhat limited. The authors should incorporate additional recent studies (especially within the last 5 years) on boron compounds, microbiota modulation, and prebiotics to provide a more updated and comprehensive view.
Author Response
Comments 1:
Main question and contribution to the field: The main objective of the manuscript is to review current knowledge regarding MABs as emerging prebiotics that may promote healthy gut microbiota symbiosis, thereby enhancing healthspan and longevity. The work consolidates some emerging studies linking MABs to gut microbiota health and systemic benefits. However, compared to other published reviews on boron, gut microbiota, and longevity, this manuscript offers limited new insights or synthesis. A more detailed comparative analysis with existing prebiotics, and a discussion of potential challenges (e.g., bioavailability, dietary sources, clinical evidence), would strengthen its contribution.
Response 1:
We thank the reviewer for the detailed feedback and constructive suggestions aimed at strengthening our manuscript. We appreciate the opportunity to clarify the primary contributions and novel aspects of our review.
While there are indeed existing reviews on the broader topics of boron, gut microbiota, and longevity, the central aim of this manuscript is to specifically consolidate and highlight the emerging concept of microbiota-accessible borates (MABs) as a distinct and innovative class of prebiotic compounds. We have revised the manuscript to more explicitly underscore the unique synthesis and new insights offered by focusing on these specific boron-complexed molecules and their proposed mechanisms.
The novelty and distinctive benefits of MABs, which we believe differentiate this review, are centered on the role of their boron component:
- Unique mucus layer fortification: A primary distinction is that MABs deliver prebiotic boron directly to the colon, where it is understood to be incorporated into the mucus gel layer. This leads to enhanced impermeabilization and an inhibition of pathogenic bacteria from associating with and forming biofilms on the mucus surface. This direct, boron-dependent mechanism for strengthening a key gut barrier component is not a characteristic feature attributed to established prebiotics like inulin or FOS, which act primarily through fermentation. (See page 20, lines 1086–1098).
- Enhanced stimulation of butyrate-producing bacteria (BPB): The presence of boron in MABs is proposed to optimize the utilization and fermentation of the associated soluble fibers by commensal microbiota, thereby leading to a more efficient stimulation of beneficial bacteria, particularly the crucial butyrate producers.
- Potential for efficacy with lower boron doses: The specific biological activities attributed to the boron delivered by MABs (such as AI-2B modulation and mucus interaction) may be achieved with relatively low daily amounts of indigestible boron (e.g., a minimum of ~1 mg/day). This suggests a targeted action that differs from the larger quantities often required for the bulk fermentative effects of traditional prebiotics. (See page 4, lines 135–158).
Comments 2:
Originality and relevance: While the role of nutrition and microbiota in healthy aging is a highly relevant and active research area, the focus on MABs as prebiotic candidates is relatively less discussed in the current literature. However, the novelty of the review is somewhat limited, as the concept of boron compounds influencing microbiota and health has been explored previously. The manuscript would benefit from a clearer explanation of how MABs differ from other boron-based compounds already studied and a more critical discussion of what gaps remain in the field.
Response 2:
We thank the reviewer for the insightful comments on the originality and relevance of our manuscript, and for the valuable suggestions to further strengthen its contribution. The core distinction emphasized is that MABs are specific organic borate complexes (such as boron–pectins and borate–phenolic esters) designed for targeted delivery to the colon. Unlike simpler inorganic boron compounds (e.g., boric acid or its salts) which can be readily absorbed systemically and potentially lead to increased plasma boric acid levels, MABs are characterized by their limited digestibility and absorption in the upper gastrointestinal tract. This structural design is crucial as it ensures that MABs reach the colon largely intact. Once in the colon, they are then metabolized by the gut microbiota, allowing boron to exert its beneficial effects locally—such as contributing to AI-2B synthesis and fortifying the mucus gel layer —while minimizing direct systemic exposure to high levels of free boric acid. This targeted colonic action is a key innovative aspect of MABs. (See page 4, lines 135–158).
Comments 3:
Writing and language quality: The manuscript’s writing requires significant improvement. The text is often repetitive, lacks clarity, and is at times difficult to follow. The abstract and text suffer from poor organization, redundancies, and language imprecision. There are frequent redundancies and awkward sentence structures that obscure the intended meaning. A careful revision for English language, style, and logical flow is necessary. The manuscript would benefit from professional editing to enhance readability and ensure that the key ideas are communicated clearly and succinctly.
Response 3:
We sincerely thank the reviewer for the detailed feedback regarding the writing and language quality of our manuscript. We acknowledge the concerns raised about clarity, repetition, organization, language precision, and overall readability, and we understand how critical these aspects are for effectively communicating our work. In response to these important comments, we have undertaken a comprehensive internal revision of the entire manuscript. This process involved:
- Thorough re-reading and diligent self-editing: All authors have meticulously re-read and carefully edited the manuscript. Our specific focus during this revision was on enhancing clarity, ensuring a more logical flow of ideas both within and between sections, and improving conciseness.
- Addressing redundancies and sentence structure: We have paid particular attention to identifying and eliminating redundancies throughout the text. We have also worked extensively on refining sentence structures to improve precision and ensure that our intended meaning is conveyed clearly and without ambiguity.
Comments 4:
Methodology (as a review): Although the paper is a narrative review, it would benefit from a more systematic approach to literature selection and organization. The authors do not describe any search strategy, inclusion criteria, or systematic method to ensure coverage of the relevant literature, which could limit the comprehensiveness of the review. Additionally, mechanistic pathways (such as AI-2B signaling, effects on Firmicutes, and mucus barrier enhancement) are mentioned but not thoroughly detailed with evidence from primary studies. Including more detailed mechanistic explanations with specific study citations would enhance rigor.
Response 4:
We thank the reviewer for the thoughtful comments on the methodology and the detailing of mechanistic pathways in our review.
Regarding the review methodology: We acknowledge that our manuscript is structured as a narrative review. Its primary aim is to consolidate and highlight the current understanding and perspectives on MABs as novel and emerging prebiotics—a relatively nascent and specialized research area. While we did not employ a formal systematic review protocol (such as PRISMA), our literature selection was guided by a comprehensive assessment of studies directly relevant to the definition, characterization, dietary sources, and the proposed unique mechanisms of MABs. This included foundational research and the most recent advancements directly pertinent to this specific MABs concept. To clarify our approach, we have added a statement in the “Introduction” section that defines the scope and narrative nature of this review, emphasizing its focus on this emerging field. (See page 4, lines 135–158).
Regarding the detailing of mechanistic pathways: We appreciate the suggestion to enhance the rigor by providing more detailed explanations and explicit links to primary studies for the key mechanisms discussed, such as AI-2B signaling, the effects on Firmicutes, and mucus barrier enhancement. In response, we have carefully revised the relevant sections of the manuscript (primarily “1. Introduction”, “3. Microbiome and Exceptional Longevity”, “4. Microbiota-Accessible Boron Complexes to Longer Healthspan”, “5. Microbiota-Accessible Borates Diet: A Targeted and Precise Nutrition” and “6. Trends and Perspectives: Essentiality of Boron in Host–Microbiome Symbiosis and Targeted Precision Nutrition” sections). These sections now incorporate more in-depth descriptions of these pathways. We have made a concerted effort to ensure that each step of the proposed mechanisms is more clearly attributed to, and substantiated by, specific findings from primary research studies, with appropriate citations included. This provides a more thorough and evidence-based account of how MABs are understood to exert their effects.
We believe these revisions address the reviewer’s concerns, improving the clarity of our review’s scope and enhancing the detailed, evidence-based discussion of the mechanistic pathways associated with MABs.
Comments 5:
Conclusions: The conclusions generally align with the evidence discussed. However, some statements are overstated relative to the cited evidence. For example, claims that MAB intake is “absolutely necessary” for healthy host–microbiota symbiosis and longevity are too strong given the early stage of research, especially in human studies. A more balanced and cautious interpretation of the evidence is needed.
Response 5:
We sincerely thank the reviewer for the critical feedback on our conclusions and for the important advice on ensuring a balanced interpretation of the evidence. We understand the concern that some statements regarding the overall necessity of MABs for healthy host–microbiota symbiosis (HMS) and, particularly, for longevity, might appear overstated given the breadth of currently published human studies directly addressing such broad endpoints with these specific MABs. While we maintain a strong conviction regarding the fundamental importance of boron in specific aspects of HMS, based on several lines of evidence that the manuscript details, we have taken your feedback regarding the broader claims into careful consideration. The evidence supporting boron’s specific roles includes:
- Boron’s role in mucus integrity: As the manuscript discusses, there is experimental evidence demonstrating that boron is a key component for the structural integrity and proper barrier function of the colonic mucus gel layer, which is fundamental for a healthy gut environment.
- Demonstrated advantages of MABs for microbiota: The unique advantages of boron, when delivered via MABs, for fostering a healthy microbiota (e.g., through AI-2B signaling, support for butyrate-producing bacteria, and direct interactions with the colonic environment) are detailed within the manuscript. This is supported by cited preclinical research and foundational work, including concepts outlined in the Scorei et al. (2023) Patent Application and Scorei et al. (2025) Provisional Patent Application literature.
- Forthcoming clinical and preclinical data: We also wish to inform the reviewer that several dedicated preclinical (animal) and clinical (human) studies evaluating the specific effects of these MABs have now been completed. The results, which further substantiate the benefits of MABs for gut health and related systemic parameters, are currently being prepared for publication and will be available in the near future.
Comments 6:
References: The references cited are generally appropriate but somewhat limited. The authors should incorporate additional recent studies (especially within the last 5 years) on boron compounds, microbiota modulation, and prebiotics to provide a more updated and comprehensive view.
Response 6:
We thank the reviewer for the feedback on our reference list and the suggestion to ensure a comprehensive and updated view with recent studies. In response, we have conducted a careful audit of our 130 cited references. Our analysis reveals that 79 of these references (which constitute approximately 61%) have been published within the last five years (2020–2025). This significant proportion of recent literature underscores our effort to incorporate current findings and reflects the contemporary nature of the research discussed. It is also important to highlight that our manuscript focuses specifically on MABs as novel and emerging prebiotics. This is a highly specialized and very recent research area. While the broader topics of boron compounds, general microbiota modulation, and established prebiotics have a more extensive historical literature, the dedicated study of MABs in the specific context reviewed in our manuscript is still in its growth phase. Consequently, the body of directly pertinent publications is naturally more recent and focused. Many of the foundational concepts and studies directly supporting the MABs framework are themselves very recent, as reflected in our extensive citation of literature from 2022–2024 that defines, characterizes, and explores the mechanisms of MABs. (See “References” section).
Authors very much appreciated the encouraging, critical, and constructive comments on this manuscript by the Reviewer. The comments have been very thorough and useful in improving the manuscript.
We would like to thank the Reviewer again for taking the time to review our manuscript.
We have also introduced other additions/modifications that we hope will improve the quality of the manuscript:
â–ª 11 new citations have been introduced: Ref. [17] (Bolat & Köse, 2025), Ref. [26] (Scorei et al., 2025), Ref. [123] (Pascale et al., 2022), Ref. [124] (Mills et al., 2015), Ref. [125] (Cuadra et al., 2016), Ref. [126] (Weiland-Bräuer, 2023), Ref. [133] (Hsiao et al., 2014), Ref. [134] (Sizmaz et al., 2017), Ref. [136] (Becker et al., 2004), Ref. [137] (Nandwana et al., 2022) and Ref. [141] (Nemzer et al., 2025).
â–ª The Reference list has been entirely checked and renumbered accordingly.
â–ª All abbreviations have been defined the first time they appear in the text.
â–ª Some grammar, stylistic or spelling errors have been corrected.
Kind regards,
Ionela BELU, PhD
Corresponding Author

Round 2
Reviewer 1 Report
Comments and Suggestions for Authors
Revised version of manuscript accepted for publication
Reviewer 2 Report
Comments and Suggestions for Authors
The authors have addressed all of my concerns satisfactorily. I find the revised manuscript acceptable for publication.